# Enterosorbents Based on Rhubarb Biomass with a Hybrid Polymer-Inorganic Coating for the Immobilization of Azaheterocyclic Mycotoxins

**Nadezhda Kornilova** [1,*], **Sergey Koksharov** [2], **Svetlana Aleeva** [2], **Olga Lepilova** [2], **Albina Bikbulatova** [3,4,*] and **Elena Nikiforova** [1]

1 Engineering Center for Textile and Light Industry, Ivanovo State Polytechnic University, Sheremetevsky Ave. 21, 153000 Ivanovo, Russia

2 Laboratory of Chemistry and Technology of Modified Fibrous Materials, G.A. Krestov Institute of Solution Chemistry, Russian Academy of Sciences, Akademicheskaya St., 1, 153045 Ivanovo, Russia; sva@isc-ras.ru (S.A.)

3 Institute of Industrial Engineering, Information Technology and Mechatronics, Moscow State University of Food Production, Volokolamskoe Highway, 11, 125080 Moscow, Russia

4 Pushchino State Natural Science Institute, Nauki Avenue, Pushchino, 3142290 Moscow, Russia

* Correspondence: nkorn@ivgpu.com (N.K.); bikbulatovaaa@mgupp.ru (A.B.); Tel.: +7-905-107-69-89 (N.K.); +7-919-727-41-80 (A.B.)

**Abstract:** The aim of the study was the improvement of the phytosorbent range to solve the actual problems of preventing mycotoxicosis caused by numerous types of azaheterocyclic mycotoxins. Technological approaches to structural released pectin and to the formation of a surface layer that was capable of adhesive interaction with montmorillonite particles was identified. The increase in the material porosity and the formation of a hybrid polymer-inorganic coating on a cellulose matrix surface were revealed by scanning electron microscopy and gas adsorption. The modification of rhubarb biomass increased sorption capacity in comparison with the raw material seven-fold. The properties of rhubarb pectin and a hybrid composite based on it were investigated using FTIR spectroscopy, viscometry, laser diffraction and X-ray diffraction analysis. The results were compared with the characteristics of commercial citrus pectin. Models of the molecular structure of the polymer chain and the spatial interaction between macromolecules in the structure of the sorbing grain were proposed based on the pectin chemical state. The influence of the pectin structural organization on the kinetic parameters of the pH-regulated sorption of the test alkaloid under conditions simulating the functioning of the human digestive organs and those of farm animals was traced. The results of the studies allow prognoses on the sorption binding of alkaloids and determinations of the dosage of pectin-containing phytopreparations for mycotoxicos prevention.

**Keywords:** phytopreparations; cellulose matrix; biomodification; self-assembled pectin–montmorillonite coating; microstructure; alkaloid absorption; kinetics

## 1. Introduction

Mycotoxins (MT) are secondary metabolites of non-pathogenic microorganisms. However, the presence of MT in feed and food products is a significant danger to human health and to the development of agroeconomics. The improvement of mycotoxicosis prevention measures is of great interest to the scientific society [1,2]. Over 400 kinds of substances secreted by microorganisms have been categorized as toxic. The negative effects of intoxication of people may include necrosis, hepatitis, haemorrhage, gynaecomastia with testicular atrophy, neurological disorders, cancer and death in extreme cases [3–5]. Similarly, animal feeds contaminated with MT can lead to reductions in available feed nutrients, chronic diseases, damage to animal health, eventual death and reduced production [6].

Regulatory restrictions of the toxigenic metabolites content in feed and food products have been introduced in most countries, including Russia [7,8]. However, mandatory control is carried out only in relation to a small group of the most common compounds. This group includes aflatoxins, ochratoxins, patulin, deoxynivalenol, zearalenone, trichothecenes and fumonisins [2–4]. The control results show a statistically low percentage of objects contaminated above permitted/guideline levels; therefore, the probability of acute intoxication is considered relatively low [9,10]. At the same time, increased attention from researchers is being paid to the analysis of the joint presence of several types of MT, the effect of which is summed up with the possibility of mutually strengthening their influence [11,12]. A multiple increase in the MT content in the environment, in residential premises, in water and food is a dangerous concomitant factor in natural disasters, such as earthquakes, hurricanes, droughts and floods [13,14].

The development of complex systems for the prevention of mycotoxicosis includes the realization of multifaceted measures aimed at both the prevention of damage and the disinfection of damaged feed and food. Strategies for prevention include controlling appropriate environmental factors, good agricultural and manufacturing practices, and favorable storage practices [15]. Strategies for MT detoxification involve the use of a variety of physical, chemical and biological influences [16–18].

The most common preventive measure of MT inactivation is the use of adsorbents. The sorbent is selected, taking into account the polarity of the neutralized compounds [19]. Increased attention is paid to achievements in dietary interventions and the mechanisms of their action against stress caused by mycotoxins. Natural dietary fibers contain various non-nutritive active molecules with significant detoxification potential and medicinal properties [20].

Methods of dietary interventions can be extremely effective for the prevention of damage by varieties of MT belonging to the group of alkaloids: nitrogen-containing heterocyclic compounds. This group has a wide composition and diverse directions of adverse effects. The formulas of some toxic alkaloids are shown in Scheme 1.

**Scheme 1.** Structure of mycotoxins structure from azaheterocyclic compounds group (structure from www.chemspider.com (accessed on 3 January 2023)): (I) ergotamine; (II) roquefortine C; (III) paxillin; (IV) β-aflatrem; (V) terpendole C; (VI) lyngbyatoxin; (VII) saxitoxin; (VIII) anatoxin-a.

Ergotamine (I) and rokuefortin (II) are the major toxins, the content levels of which are regulated in feed and food products. Review [21] provides comprehensive information about a large group of recently identified indole-diterpenes of neurotoxic and tremorgenic action. These include paxillin (III), β-aflatrem (IV) and terpendol C (V), which induce

neurologic symptoms ranging from mental confusion to tremors, seizures and death. A number of neurologic diseases of cattle collectively known as staggers syndromes are also related to their influence.

The enumeration will be incomplete if the widespread group of alkaloids produced by toxigenic cyanobacteria during the flowering of fresh and marine water reservoir (blue-green algae) are not mentioned [22]. Lyngbyatoxin (VI) belongs to the dermatotoxins group and provokes the formation of malignant tumors. Saxitoxin (VII) and anatoxin-a (VIII) can accumulate in fish and shellfish bodies. They cause toxic effects if ingested in organisms, even in nanomolar concentrations [23,24].

Comparison of the formulas of the listed MT (see Scheme 1) demonstrates that the use of preliminary adsorption binding of alkaloids methods is futile. The molecules of such compounds acquire a positive charge when ingested in the strongly acidic medium of the stomach as a result of the protonation of nitrogen atoms. This provides them solubility, diffusion mobility and the possibility of unhindered absorption into the walls of the thin intestine.

Pectin-containing plant sorbents are an effective means of binding protonated compounds. This was demonstrated, in particular, by the example of cleaning industrial wastewater from azo dyes [25,26] and also in the development of perspective dietary supplements to reduce the negative effect of MT on the work of the internal organs of animals and humans [27,28]. The resistance of pectin substances to enzymolysis causes their transit passage through the digestive system (i.e., not digested). It can prevent or minimize the harmful effects of adsorbed MT.

The difficulty of immobilization protonating MT is that pectins are inactive in a highly acidic medium of the stomach (pH 1.5–2.5) due to the suppression of carboxyl groups' dissociation. At the same time, pectins become active when they enter the duodenum. The pH level is 4.5 units at the entrance to the duodenum, and at the exit it is 6.8 units. Apparently, pectins can prevent the absorption of alkaloids into the walls of the thin intestine if they provide a strong binding of MT during the 20-min passage of food through the human duodenum. The length of the duodenum in ruminants differs from the size of a human organ (50 cm in goats and sheep, 120 cm in cows and up to 200 cm in bulls and buffaloes). The duration of the passage of feed through this part of the cow's intestine is about 30 min.

In previous studies, we have shown the existence of a relationship between the features of the chemical structure of pectins' different plant nature and their sorption capacity regarding cationic pollutants [29,30]. Approaches for the estimation of kinetic parameters of the stages of diffusion and chemisorption interaction of azaheterocyclic MT with flax-containing feed additives for farm animals have been proposed [31]. The regularities for substantiating the consumption rates of enterosorbents from flax seed husks in comparison with commercial apple pectin samples have been revealed [32].

The assessment of the preventive ability of the herbaceous plant rhubarb (genus *Rhéum*) biomass is of practical interest because it contains various biologically active substances including pectin and is widely uses in phytotherapy [33]. Earlier [34], we found that, in order to increase the adsorption capacity of pectin, its release from the polymer environment in the material structure is necessary. The biomodification of plant raw materials allows an increase in the cellulose amorphousness and ensures an almost complete (89%–97%) release of pectin substances from the cellulose matrix.

We took into account the increased interest of researchers in the use of aluminosilicate sorbents for the prevention of mycotoxicosis [35]. An actual task for expanding the range of their application is the increase of strength of the sorption binding of toxins in cationic form. The prospects of this direction can be enhanced thanks to the development of methods for simulating the alkaloids' sorption by binary polymer-inorganic systems. The prerequisites for an in-depth study of these processes are based on the previously established [36,37] strengthening of the mutual action of the components of pectin–montmorillonite complexes to inorganic and organic pollutants. However, there is no information available to prognos-

ticate the effectiveness of using hybrid composites to ensure the pH-regulated binding of mycotoxins during the passage of sections of the digestive system. The identification of the regularities of the structural organization polyuronides' influence on their interaction with montmorillonite will allow us to substantiate the methods of obtaining enterosorbents and the conditions for their practical application.

The purpose of this study was to evaluate the effectiveness of plant raw materials' biochemical modification to form a sorption-active pectin–montmorillonite coating and to identify the "composition-property" patterns for the description of the sorption kinetics of theophylline, taking into account the data on the structural features of rhubarb polyuronides for the simulation of the binding processes of azaheterocyclic MT in the bodies of humans and farm animals.

## 2. Materials and Methods

### 2.1. Materials

The model of azaheterocyclic MT was used a compound, as presented in Scheme 2. It is a system of pyrimidine and imidazole cycles with two common carbon atoms, belonging to the group of purine bases.

**Scheme 2.** Formula of 1,3-dimethyl-7H-purine-2,6-dione (theophylline).

The trivial name of the compound is theophylline. It is applied as a pesticide and a medicine. According to the value of the median lethal dose $LD_{50}$ index (225 mg·kg$^{-1}$), the substance belongs to the third hazard class (moderately hazardous substances) which allows it to be used in experimental chemical research. The pharmaceutical form of the drug was used in the study (manufactured by Valenta Pharm JSC, Moscow, Russia).

In the study, the biomass of the garden culture of rhubarb *Rhéum rhabarbarum* L. (Rh), which grows in the central regions of Russia and belongs to the edible plants category, was used as a raw material. Initially, the raw material was washed in warm (40 °C) water, dried at room temperature and crushed to a particle size of about 1 cm.

Commercial cellulase preparation Xybeten®-Cel (Biovet JSC, Sofia, Bulgaria) was used for Rh biomodification. Xybeten®-Cel contains cellulase (endo-β-1.4-glucanase) with an activity of 15,000 U·g$^{-1}$, beta-glucanase (endo-β-1,3-glucanase) with an activity of 5000 U·g$^{-1}$ and xylanase (endo-β-1,4-xylanase) with an activity of 150 U·g$^{-1}$.

The objects of the study are designated as indicated in Table 1.

**Table 1.** Objects of research.

| Symbol | Preparation |
| --- | --- |
| Rh–Mt | hybrid rhubarb–montmorillonite phytocomposites based on the original Rh biomass |
| Rh$^{bio}$–Mt | hybrid rhubarb–montmorillonite phytocomposites based on the biomodified Rh biomass |
| P$_{Rh}$ | pectin preparation isolated from biomodified Rh biomass |
| P$_{Ct}$ | commercial citrus pectin preparation (reference sample) |
| P$_{Rh}$–Mt | hybrid nanocomposites of the compared pectins from Rh biomass with montmorillonite |
| P$_{Ct}$–Mt | hybrid nanocomposites of the compared citrus pectin with montmorillonite |

The Mt was purchased from Sigma Aldrich (St. Louis, MO, USA). Pectin $P_{Ct}$ was acquired from Yantai DSM Andre Pectin Co., Ltd. (Yantai, China). All reagents used for the isolation and analysis of the obtained samples were of analytical grade and were purchased from Sigma Aldrich (St. Louis, MO, USA) and Fisher Scientific (Montreal, WI, USA).

### 2.2. Rhéum Rhabarbarum Preparation

Rh biomodification was carried out using low-modulus processing technology. A total of 50 g of the substrate was mixed with 250 mL of the enzyme preparation solution with cellulase activity of 75 U·mL$^{-1}$, beta-glucanase 25 U·mL$^{-1}$ and xylanase 0.75 U·mL$^{-1}$ at pH 5.4. The mixture was heated and kept at $40 \pm 5$ °C (optimal temperature of enzymes) for 2 h at constant stirring. Then, the mixture was dried without flushing at 80 °C until the enzyme was inactivated.

To extract $P_{Rh}$, 10 g of the $Rh^{bio}$ preparation crushed in a mortar was mixed with 250 mL of distilled water. Then, a cold ultrasound treatment using the UZDN-2T disintegrator (LLC U-RosPribor, Moscow, Russia) was carried out at a frequency of 22 kHz for 5 min. Ultrasound treatment was carried out in an ice bath. The temperature of the hydrosol during extraction was not higher than 50 °C. The filtrate was precipitated with 96% ethyl alcohol in a ratio of 1:2 (*v/v*) for obtaining the $P_{Rh}$ preparation. The coagulate was filtered through a pre-weighted Miracloth. The separated coagulate was washed with 70 and 96% ethyl alcohol, and dried at 60 °C for 24 h.

### 2.3. $P_{Rh}$–Mt and $P_{Ct}$–Mt Preparation

To obtain the bicomponent $P_{Rh}$–Mt and $P_{Ct}$–Mt systems, the method for producing montmorillonite–starch composites was adapted [38]. Montmorillonite slurry was previously prepared with distilled water for swelling (Mt/water ratio was 70:30). Next, a weighed sample of the powder pectin was mixed with the swollen montmorillonite in a ratio of 90:10 using the vibration mill (operating parameters: vibration frequency of the activator was 50 Hz, vibration amplitude was 180 µm, vibration velocity was 158 mm·s$^{-1}$, vibration acceleration was 140 m·s$^{-2}$ and the exposure duration was 30 min). The obtained composite was dried at a temperature of 40 °C for 1 h (vacuum drying at 66.5 kPa) with further exposure in a desiccator with anhydrous calcium chloride.

### 2.4. Rh–Mt and $Rh^{bio}$–Mt Preparation

The Rh–Mt and $Rh^{bio}$–Mt bioactive hybrid-nanocomposites were prepared by the treatment of dried plant samples with working suspensions of Mt at 50 °C for 2 h. The amount of Mt added into the biomass ($G_{Mt}$) was varied from 2 to 50 wt.%. The obtained samples were dried at 105 °C for 2 h. The efficiency of the Mt immobilization into the biomass structure was determined by the washout of the aluminosilicate filler during treatment with distilled water for 20 min without stirring, or for 5 min with stirring using a blade stirrer at 30 rpm. The plant biomass was quantitatively separated from the washing liquid using a cloth filter, and dried. After that, the change in the weight of the separated mass compared to the initial sample was determined The calculation of the residual Mt content on the biomass $G_{res}$ (wt.%) was carried out using the formula $G_{res} = G_{Mt} - \left( G_{Rh-Mt}^{source} - G_{Rh-Mt}^{wash} \right)$, where $G_{Rh-Mt}^{source}$ and $G_{Rh-Mt}^{wash}$ are the weight of initial and after-washing hybrid nanocomposites. The error rate of $G_{res}$ determination was = 0.5 wt.%.

### 2.5. Evaluation of the Rh–Mt and $Rh^{bio}$–Mt Structure

The state of the pore Rh–Mt and $Rh^{bio}$–Mt structures and their components was evaluated using the low-temperature nitrogen adsorption–desorption method at 77 K on a NOVA 1200e gas sorption analyzer. A 0.25 g sample was placed in a cell and its degassing was carried out under vacuum for 8 h at 100 °C. After weighing, nitrogen was injected into the cell with a sample. The nitrogen molecules condensed on the sample surface, forming a monolayer. For a hexagonal dense nitrogen monolayer at 77 K, the calculated value of the



N molecular cross-sectional area was 16.2 Å$^2$. Measurements were carried out in the range 0.01–1.0 units of relative pressure P/P$_o$. The pore volume V$_P$ (cm$^3$·g$^{-1}$) and the specific surface area of materials S$_A$ (m$^2$·g$^{-1}$) were calculated with the Brumauer–Emmett–Teller (BET) method using the NOVA Win-2.1 software.

Scanning Electron Microscopy (SEM) (Quattro S, Thermo Fisher Scientific, Brno, Czechia) was used for the observation of the phytomaterials' surface.

### 2.6. Evaluation of the Pectin Structure

Bruker Vertex 80 V (Ettlingen, Germany) was used to identify pectins using FTIR spectroscopy. The FTIR spectra of the pectin were recorded in the region of 400–4000 cm$^{-1}$ from pellets with KBr. The concentration of the samples in the pellets was constant at 3 mg/300 mg KBr. The spectra were baseline corrected for further analysis.

The quantitative determination of the content in the pectin of non-substituted (***N***), methoxylated (***M***) and calcium–pectate (***C***) galacturonate units was determined by FTIR spectroscopy of polymer films in accordance with the previously described experiment procedure [29,39]. The method is based on using the isolated band at 1620 cm$^{-1}$, which is formed by valence vibrations of the bond of the carboxyl group with metal ions $\nu_{as}$(C–OMe) [40]. The technique of the preparation of auxiliary samples includes cycle treatments of pectin films for their sequential conversion into pectinate (a mixture of ***M***- and ***C***-forms), polygalacturonic acid (***N***–form) and calcium pectate (***C***–form). The change in the intensity of the band at 1620 cm$^{-1}$ in the pectin spectra and auxiliary substrates was estimated with respect to the band of the internal standard, which was formed by valence vibrations between the atoms of the pyranose cycle of $\nu_{as}$(C–C) at 1020 cm$^{-1}$. The FTIR spectra were taken in the transmission mode in the region of 500–4000 cm$^{-1}$ using the AVATAR-360 spectrophotometer.

The determination the molecular weight of pectins was carried out with the viscosimetric method using a thermostatically controlled Ubbelohde viscosimeter. The presence in the pectin of intermolecular crosslinking formed by calcium–pectate units necessitated the preliminary decalcification of biopolymer preparations. The treatment of P$_{Rh}$ and P$_{Ct}$ pectins was carried out using a sodium ethylenediaminetetraacetate (EDTA) solution in 70% ethanol at 25 °C for 1 h. The experimental evaluation included a standard procedure of measuring the kinematic viscosity (η) of 0.1%–1% pectin hydrogels and the construction of graphical dependencies $\eta_{sp}/c = f(c)$ and $\ln[(\eta/\eta_0)/c] = f(c)$ for the determination of the intrinsic viscosity.

The molecular weight of pectin ($M_P$) was calculated using the Mark–Kuhn–Houwink equation $[\eta] = kM_P^a$, where $k$ and $a$ are coefficients characterizing the shape of macromolecule and polymer–solvent interactions ($a$ = 1.22; $k$ = 1.1 × 10$^{-5}$).

The degree of polymerization (*DP*) was determined as the ratio of the molecular weight of pectin to the molecular weight of the monomeric galacturonate unit ($M_{GA}$ = 192 g·mol$^{-1}$) according to the formula $DP = M_P/M_{GA}$.

### 2.7. Evaluation of the P$_{Rh}$–Mt and P$_{Ct}$–Mt Structure

X-ray diffraction analysis with a DRON-3 diffractometer was used to determine the interlayer spacing in the Mt structure.

The estimation of the particle size in the dispersed samples was carried out with laser diffraction with an Analysette 22 Compact Analyzer (Compact Fritsch, Idar-Oberstein, Germany).

### 2.8. Kinetics of Theophylline Sorption

The theophylline binding was analyzed using a classical method of sorption from a limited volume in static conditions. Thermostating at 40 °C was used to simulate the physiological modes of food digestion. Phosphate buffer solutions were used for achieving pH 2.0 and 5.0. Kinetic sorption curves were obtained with a series of experiments, with variations of the treatment duration of the pectin sample in the theophylline solution. A

total of 20 mL of theophylline solution with an initial concentration ($C_0$) of 2.5 mmol·L$^{-1}$ was placed in thermostatic flasks and mixed with 5 mL of 0.4% pectin solution. Sampling for analysis was carried out every 5 min with a total duration of the experiment from 60 to 180 min. Before analysis, the samples were centrifuged for 10 min at 2000 rpm to separate the sorbent. The change in the concentration of theophylline in the supernatant during the sorption experiment $t$ ($C_t$) was fixed using a spectrophotometer at a wavelength of 270 nm. The calculation of the current values of theophylline sorption was carried out according to the formula $q_t = (C_0 - C_t) \cdot V / (m \cdot M_T)$, where $V$ is solution volume, $m$ is the weight of the sorbent (g) and $M_T$ = 180 g·mol$^{-1}$ is the molecular weight of theophylline.

## 3. Results and Discussion

### 3.1. Structure and Sorption Properties of the Modified Phytocomposite

Figure 1 demonstrates the model of the complex structure of the cell wall plants. The model demonstrates that the pectin role is the bonding of cellulose microfibrils and adjoined macromolecules of hemicellulose compounds passing from one cellulose bundle to another and filling the interfibrillary space. The three-dimensional structure of the carbohydrate–protein complex is fixed by point interactions between the carboxyl groups of pectins and amino groups of orthogonally located protein macromolecules.

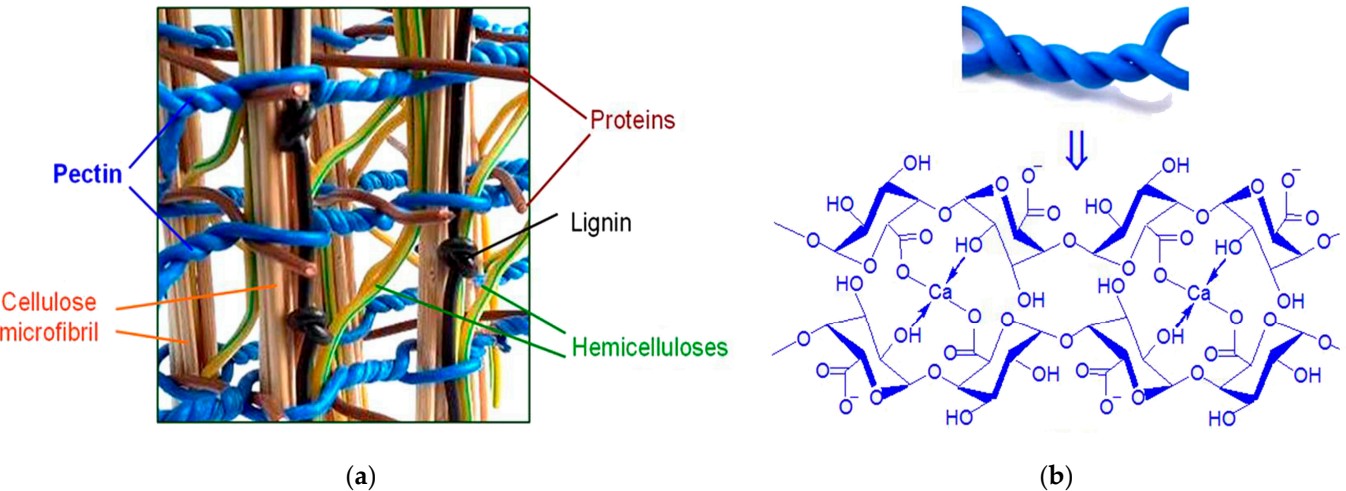

(**a**)  (**b**)

**Figure 1.** Schemes (**a**): Carbohydrate–protein complex structure in the plant cell wall and (**b**): Interchain crosslinking of pectin macromolecules by "egg-box" complexes.

Rh contains 12.5% of pectin out of the dry matter weight. Ultrasonic treatment allows the extraction of no more than 2.5% of pectin from the initial raw materials. The preliminary effect of the enzyme composition ensures the transverse destruction of hemicelluloses and cellulose microfibrils, contributing to the release of pectin substances. At that, the mesh structure of pectins is preserved with the alternation of cells and "egg-box" complexes. Cells consist of fragment macromolecules, earlier braiding the cellulose macrofibrils. "Egg-box" complexes are inter-molecular crosslinkings with the participation of calcium ions (see Figure 1b) [41].

The yield of pectin during Rh$^{bio}$ extraction increased 4.6 times. At that, the low-modulus biomodification of raw materials without washing allowed the preservation of pectin in the material structure. The structural release of pectin is a favorable factor for the subsequent mixing of the phytopreparation with a moistened Mt dispersion.

Figure 2 demonstrates the results of Mt immobilization analysis during the treatment of initial and biomodified plant raw materials.

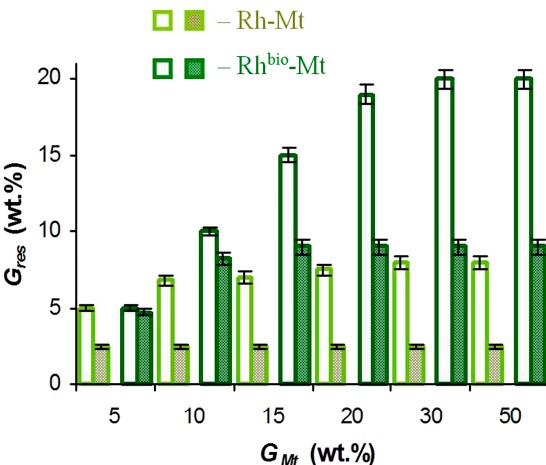

**Figure 2.** The effect of the amount of montmorillonite injected into the biomass and subsequent washing without stirring (without hatching) or with stirring at 30 rpm (hatching) on the weight gain of bicomponent samples.

It was found that Mt can be mechanically retained in the biomass, which is also due to adhesive interactions. It was differentiated by changing the washing mode of the bicomponent samples. The amount of firmly fixed Mt in the Rh$^{bio}$ structure was about 9 wt.%. The part of Mt in excess of 20 wt.% desorbs into the without-stirring washing. Therefore, this part did not have even a mechanical binding with the biomodified plant substrate. The total amount of mineral filler retained in the initial raw material structure did not exceed 8 wt.%, and 2.5 wt.% of this was not removed from the Rh-Mt sample under washing with mechanical stirring; that is a fact of a strong adhesive bond with the substrate.

The change in the pore structure parameters of the used materials and bicomponent samples depending on the montmorillonite proportion is characterized by data presented in Table 2.

**Table 2.** Characteristics of the pore structure of hybrid phytocomposites and their components.

| Ratio of Rh:Mt in the Sample | $V_P$ (cm$^3$·g$^{-1}$) | | $S_A$ (m$^2$·g$^{-1}$) | |
|---|---|---|---|---|
| | **Rh–Mt** | **Rh$^{bio}$–Mt** | **Rh–Mt** | **Rh$^{bio}$–Mt** |
| 100:0 | 0.019 | 0.070 | 14.0 | 57.2 |
| 97.5:2.5 | 0.027 | 0.078 | 32.8 | 75.0 |
| 95:5 | 0.034 | 0.089 | 49.7 | 93.8 |
| 90:10 | 0.045 | 0.102 | 83.6 | 132.6 |
| 0:100 | 0.270 | | 690.8 | |

The biomodification of plant raw materials provides an increase in the pore volume $V_P$ by 3.7 times. The pore-specific surface area of $S_A$ increases by 4.1 times. Rh–Mt and Rh$^{bio}$–Mt samples containing 2.5% of Mt have an equal increase in porosity, which reflects the identity of the state of the layered aluminosilicate. At that, a further increase in the mass fraction of Mt contributes to a more intensive increase in the absolute values of $V_P$ and $S_A$ for Rh$^{bio}$–Mt photocomposites. Consequently, adhesion more effectively changes the Mt state in the biomodified material. At that, the experimental values of porosity indicators exceed the calculated additive contribution of the system components, and the final result of the modification provides an almost 10-fold increase in the sorption-active surface in comparison with the raw material.

The SEM images presented in Figure 3 allow the fixing of the change in the state of the plant material surface at successive stages of obtaining phytocomposite. The particles of the initial dried raw materials have the uneven surface characteristic of spongy plant tissues. The surface becomes smoother after the biomodification stage. This result is

due to an increase in the migration ability of structurally released pectin and its transfer onto the surface of moisture evaporation at drying of the biomodified materials. Pectin has a high hydrophilicity and actively absorbs moisture when subsequently mixed with swollen layers of aluminosilicate. The layer of pectin hydrocolloid forms becomes capable of adhesive bonding with the Mt particles, and forms a hybrid polymer-inorganic coating on the cellulose matrix surface [36,37].

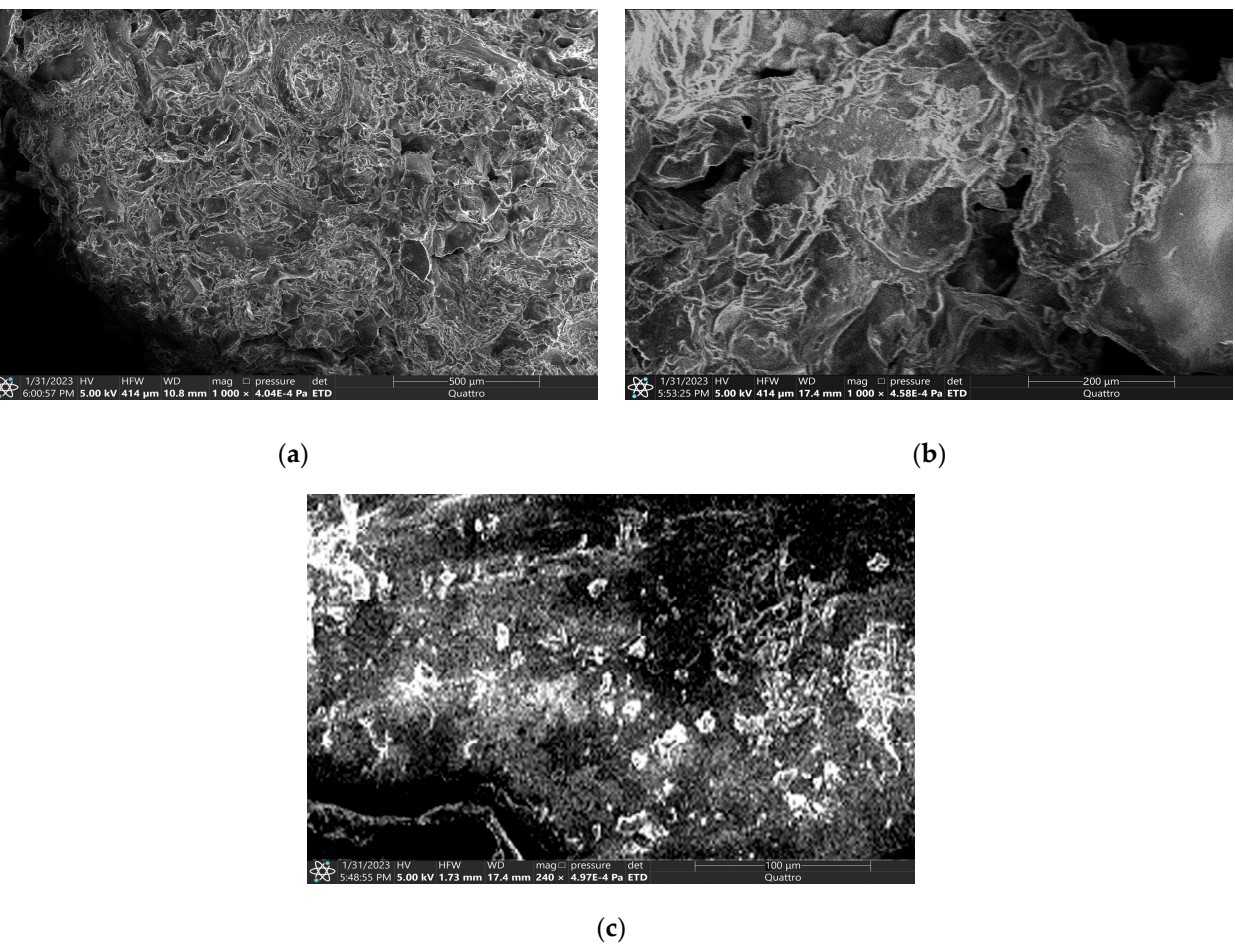

**Figure 3.** SEM images of the surfaces: (**a**) Rh; (**b**) Rh$^{bio}$; (**c**) Rh$^{bio}$–Mt.

The results of the sorption capacity evaluation of raw materials, intermediates and modified phytocomposite regarding the model alkaloid are given in Table 3. The selected time of the sorption experiment (*t*) corresponds to the passage duration of the food in the human duodenum (20 min) and that of large ruminants (30 min), as well as the period of reaching sorption equilibrium (180 min for the initial Rh and 60 min for biomodified raw materials and phytocomposites).

**Table 3.** Theophylline sorption by initial and modified substrates (40 °C, pH 5).

| Sorption Time, *t* (min) | Quantity of Sorbed Theophylline, $q_t$ (mg·g$^{-1}$) | | | |
|---|---|---|---|---|
| | Rh | Rh$^{bio}$ | Rh–Mt (90:10) | Rh$^{bio}$–Mt (90:10) |
| 20 | 0.90 | 18.38 | 8.83 | 39.24 |
| 30 | 1.44 | 18.92 | 9.91 | 39.52 |
| 60; (180) | (5.59) | 18.92 | 10.45 | 39.91 |

The release of pectin from the Rh$^{bio}$ cellulose matrix allows an increase of the equilibrium absorption of theophylline ($q_e$) by 3.4 times. At that, the initial rate of alkaloid binding increases by 20.4 times, allowing the $q_t$ value to have already reached 97% of its level at the first control time point. Pectin goes out onto the mesopore material surface and facilitates its interaction with theophylline molecules, as well as with the particles of the applied Mt dispersion.

The application of a mineral component onto an unmodified plant substrate also accelerates the sorption binding of theophylline. However, the level of equilibrium absorption increases by less than two times. We can assume that the fiber and the Mt presenting in the phytocomposite work individually, providing an additive sorption result.

The experimental results for the Rh$^{bio}$–Mt sample indicate that a synergism in the action of the vegetable fiber components and the applied mineral filler is expressed. The sorption properties of the system are almost fully realized within 20 min. As shown above, this time is necessary for the effective binding of mycotoxins in the duodenal zone of humans. At that, the value of the equilibrium sorption capacity of the modified phytocomposite exceeds the $q_e$ level for initial raw by 7.1 times. The significance of the observed changes allows the suggestion that a hybrid pectin–montmorillonite layer of composite was formed on the material surface. The interest in studying the interaction between biomodified rhubarb pectin and Mt is also due to the possibility of direct P$_{Rh}$–Mt preparation use as a specialized enterosorbent for the prevention of mycotoxicosis.

### 3.2. Structure and Sorption Properties of the Hybrid Polymer-Inorganic P$_{Rh}$–Mt Coating

The simulation of the state and the sorption behavior of the surface layer onto the fibrillar basis of the Rh$^{bio}$–Mt sample were carried out using P$_{Rh}$ pectin isolated from biomodified Rh biomass. Figure 4 shows the identification results of the P$_{Rh}$ sample in comparison with the commercial citrus pectin (P$_{Ct}$).

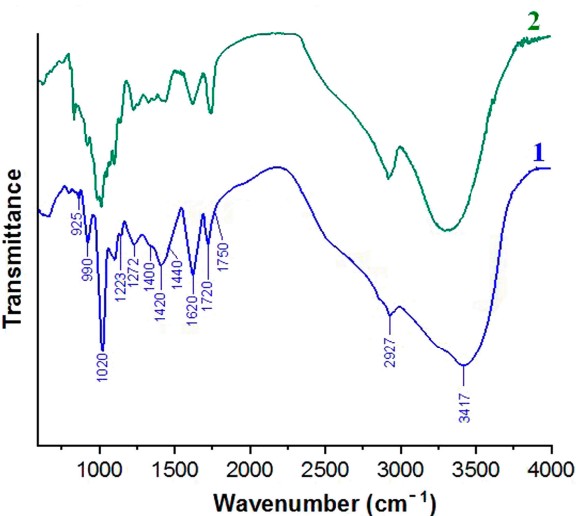

**Figure 4.** FTIR spectra of P$_{Rh}$ (1) and P$_{Ct}$ (2) pectins.

The peak positions of the predominant types of valence vibrations in the analyzed samples correspond with the data given in [40] (pp. 269–336). This confirms the high degree of purity of the P$_{Rh}$ preparation. The typical valence vibrations in the covalent bonds of pectin macromolecules are:

- 1750 cm$^{-1}$—the valence vibrations of carbonyls in ester groups $\nu_{as}$(C=O);
- 1720 cm$^{-1}$—the vibrations of C=O bonds in non-esterified carboxyl groups;
- 1420 and 1615 cm$^{-1}$—the symmetric and asymmetric valence vibrations of carboxyl with metal cations $\nu_s$, $\nu_{as}$(C–OMe);
- 1440 cm$^{-1}$—the deformation asymmetric vibrations in the methoxyl group $\delta_{as}$(O–CH$_3$);

- $1400$ cm$^{-1}$—the valence vibrations of the C-OH bond in carboxyl $\nu$(C–OH);
- $1272$ and $1223$ cm$^{-1}$—the valence vibrations of ester bond $\nu$(C–O–C);
- $1020$–$1010$ cm$^{-1}$—the valence vibrations of pyranose rings $\nu$(C–C)(C–O);
- $990$ cm$^{-1}$—the deformation vibrations of carboxyl $\delta$(C–OMe);
- $920$ cm$^{-1}$—the pendular vibrations of methyl in the ester group $\rho$(O–CH$_3$).

The direct determination of the content of methoxylated and non-substituted carboxyl groups in the polymer is difficult because the isolation of the individual vibration bands from the complex superposition of the overlapping vibration processes in the frequency range of $1700$–$1750$ cm$^{-1}$ and $1350$–$1450$ cm$^{-1}$ is impossible. Analysis of the methoxyl group number by the absorption bands maxima of the valence vibrations of the ester bond $\nu$ (C–O–C) at $1272$ and $1223$ cm$^{-1}$ is also impossible, because in this region the absorption bands of the deformation vibrations C–H bond of pyranose rings and O–H bond of the alcohol hydroxyl group overlap [29,40].

The peak at $1620$ cm$^{-1}$ is an isolated band of the asymmetric stretching vibrations of the C–O bond in the pectate. The assessment of the intensity change of this stretching vibrations band forms the basis for the method we applied to determine the fractional content of galacturonate units in the unsubstituted, methoxylated and calcium–pectate forms. The method is implemented using the above successive transformations of the free (non-esterified) and methoxylated forms of galacturonic acid into calcium pectate, with the registration of the specific absorption increase at $1620$ cm$^{-1}$ by the method of FTIR spectroscopy of the polymer films.

Table 4 compares the pectin preparations under study by the content of galacturonate units with carboxyl groups in the non-substituted (**N**), methoxylated (**M**) and calcium–pectate (**C**) forms. The results are presented as a fraction of the total number of galacturonate units.

**Table 4.** Analysis results of the chemical structure of pectin preparations.

| Pectin | Fractional Content of Galacturonate Units Forms (Units) | | |
| --- | --- | --- | --- |
| | COOH  Symbols: *N*; 🟡 | COOCH$_3$  Symbols: *M*; 🟢 | COO–Ca  Symbols: *C*; 🔴 |
| P$_{Rh}$ | 0.28 | 0.53 | 0.19 |
| P$_{Ct}$ | 0.15 | 0.65 | 0.20 |

Both preparations are highly methoxyl pectins in which more than 50% of **M**-units. Those preparations have a close ratio of units in the C-form, but the content of the sorption-active form with unsubstituted carboxyl groups differs almost twofold.

The presence of a large number of C-units in the preparations necessitates the preliminary decalcification of pectin to experimentally determine its molecular weight ($M_P$) and degree of polymerization (*DP*). According to [42], the pectin molecular weight can be $6.91 \times 10^3$ kDa. In our opinion, it is a mistake to consider this as characteristic of an individual macromolecule because it is necessary to take into account the presence of interchain crosslinking "egg-box" (see Figure 1b) between macromolecules. It is known that macromolecules of cotton cellulose, having a degree of polymerization an order of magnitude higher than that of wood cellulose, have a maximum value of molecular weight of only $2.43 \times 10^3$ kDa [43]. Therefore, the $M_P$ values at the level of $615.8$ kDa [44] and even $488$ kDa [45] found in the literature are not characteristic of the separate macromolecular chain, and can only reflect the state of associates that are bonded by intermolecular crosslinking.

The applied method of calcium ion extraction using sodium ethylenediaminetetraacetate is based on the fact that the stability of the CaEDTA complex (lg $K = 10.56$) is nine orders of magnitude higher than the stability of calcium galacturonate (lg $K = 1.84$) [46].

The comparison of characteristics for samples before and after decalcification (Table 5) indicates that $P_{Rh}$ associates consist of six to seven macromolecules. The $P_{Ct}$ preparation contains associates of four to five macromolecules, having almost two times shorter length in comparison to native pectin $P_{Rh}$. Apparently, lower values for the $P_{Ct}$ are associated with the partial polymer destruction during extraction and the production of a commercial form of the preparation.

**Table 5.** Results of the viscosimetric assessment of the molecular weight of pectin preparations.

| Pectin | Decalcification | $[\eta] \pm 0.05$ $(cm^3 \cdot g^{-1})$ | $M_P \pm 0.1$ (kDa) | $DP \pm 5$ |
|---|---|---|---|---|
| $P_{Rh}$ | − | 23.7 | 155.6 | 802 |
| | + | 2.3 | 23.3 | 120 |
| $P_{Ct}$ | − | 7.4 | 60.7 | 313 |
| | + | 1.15 | 13.0 | 68 |

The obtained physico-chemical pectin preparation's characteristics make it possible to simulate the block-cellular structure of associates, which is based on the following initial postulates. The galacturonate unit in the *C*-form cannot be located individually as part of a flexible segment because the $Ca^{2+}$ interacts simultaneously with two carboxyl groups in neighboring macromolecules and the calcium bridges are formed. Single intermolecular crosslinking will be unstable. Stable conformational formation of the "egg-box" occurs only in the presence of several similar calcium bridges in neighboring units [41]. The number of groups crosslinking in the "egg-box" blocks of natural pectin, as a rule, does not exceed four [47], and the minimum number is two (see Figure 1b). An individual non-substituted unit cannot sorb the $Ca^{2+}$ ion and is present in the composition of flexible branches only when surrounded by methoxylated units (see Scheme 3).

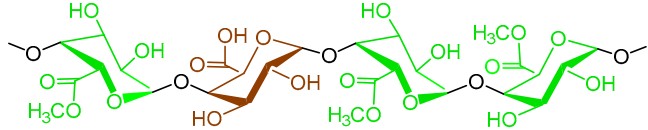

**Scheme 3.** Alternation of galacturonate units in flexible segments of the mesh structure of pectin (*M*–form is marked by green color; *N*–form is marked by brown color).

Based on the above postulates, we modeled the most probable mutual arrangement of structural subunits of pectins in accordance with the data of Tables 4 and 5:

- The total composition of units for the $P_{Rh}$ sample at $DP = 120$ is $C_{24}N_{32}M_{64}$ and has the distribution $[(C_3N_2)(M_8N_2)]_8$. Each macromolecule passes through eight blocks, which are formed by three *C*-links and between them two links in the *N*-form. Blocks are separated by branches consisting of eight *M*-units with two imbedded *N*-units.
- The total composition of units for the $P_C$ sample at $DP = 68$ is $C_{16}N_{12}M_{40}$ and the distribution of subunits is $[(C_2N_2)M_5+(C_2N_1)M_5)]_4$. Each macromolecule passes a paired combination of blocks four times, which collectively contain four units in *C*-form and three units in *N*-form; that is, they are formed with the participation of a double alternation of *C*–*N* units from one chain and a fragmentation of *C*–*N*–*C* from an adjacent chain. The branches between the blocks contain five units in *M*-form.

In Table 4, graphic symbols are assigned to galacturonate units. They were used to form the scheme illustrated in Figure 5 that shows models of the statistically probable formation of crosslinking blocks and a branched part in the chain of main valences of the studied pectin samples.

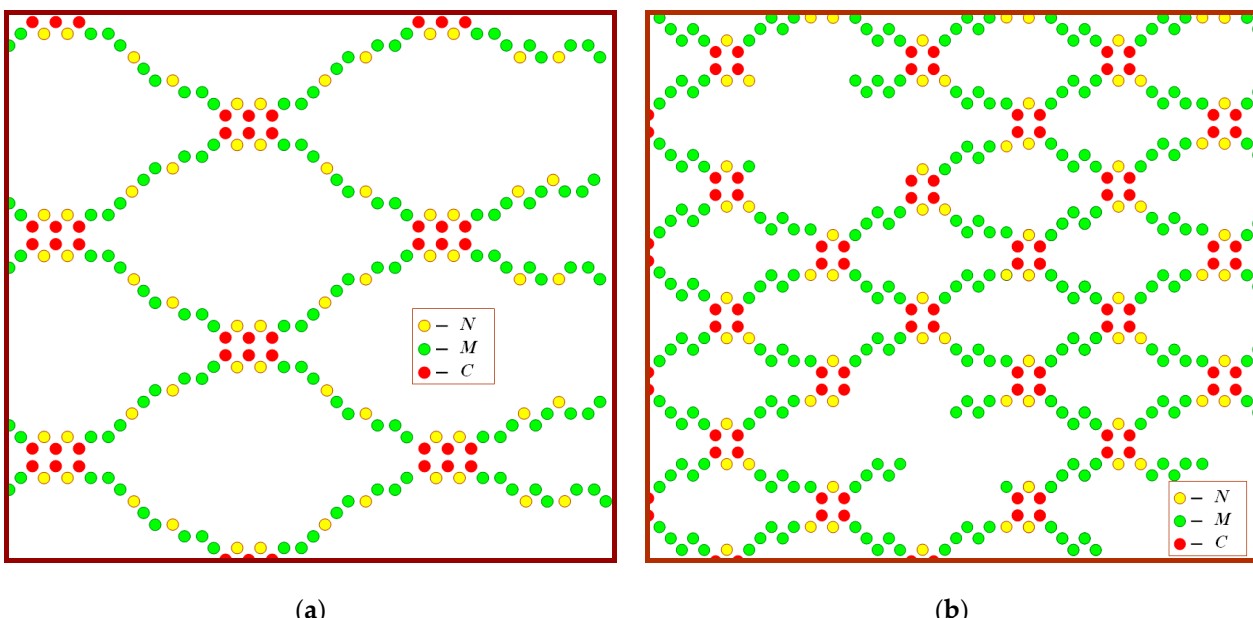

**(a)**                                                                                      **(b)**

**Figure 5.** Mesh structure of pectin: (**a**) $P_{Rh}$ sample; (**b**) $P_{Ct}$ sample.

The following assumptions are made when constructing the models. First of all, they are related to the two-dimensional representation of the macromolecular chains interconnections, simplifying the actual 3D arrangement of the polymer meshes. This version can be considered as a fragment of a single mesh layer, from which macromolecules can move to the adjacent parallel layers. To simplify, the number of *C*-units in the crosslinking blocks is assumed to be constant and multiple that of the experimental value of the content of this form in the polymer. The number of monomer units in flexible segments between seals is also proportionally distributed. Finally, in the accepted version, the polymer structures are formed by homogalacturonan macromolecules. For real pectin substances, 2%–4% of $\alpha$-rhamnose residues have to be included into the diagrams [48]. The lateral branches from the rhamnose units will be oriented in the orthogonal direction, in parallel with the location of cellulose fibrils and hemicelluloses (see Figure 1b), so they do not affect the formation of the crosslinked mesh structure.

The schemes demonstrate the difference between objects disappearing behind the numerical ratio of the forms of galacturonic units (Table 4). It is obvious that the deficiency of the *N*-form in the $P_{Ct}$ preparation causes a shorter length in the sites involved in the "egg-box" blocks. The length of flexible domains is reduced simultaneously, affecting the swelling of the polymer in water.

The high openwork nature of branched domains in the $P_{Rh}$ sample determines the hyperelastic and textural properties of rhubarb gum gels noted by specialists [49]. It is this state in which polyuronide functions in the digestive system upon intake of the analyzed pectin containing preparations.

It should be noted that the procedure of the solid-phase preparation of model $P_{Rh}$-Mt and $P_{Ct}$-Mt samples was maximally similar to the conditions of treatment of $Rh^{bio}$ biomass with the layered mineral. The preliminary saturation of Mt with water ensures the mobility of its layered structure. On the other hand, a limited amount of moisture redistributed in the system is insufficient for the complete hydration of the pectin phase. Therefore, as in the phytomaterials, the possibilities of mutual penetration of the phases are limited, which is important for preserving the internal volume of the Mt interlayer spaces in the active state.

Figure 6 shows the change in size parameters of the dispersed phase of the interacting components for the $P_{Rh}$-Mt sample with 10% Mt content.

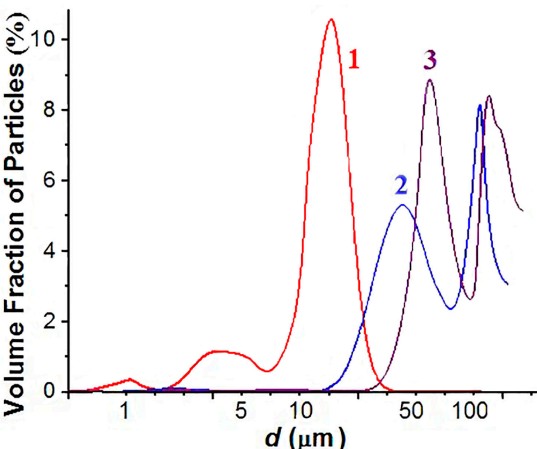

**Figure 6.** Differential particle-size distribution curves of samples of (1) Mt, (2) $P_{Rh}$ and (3) $P_{Rh}$-Mt systems (90:10).

It was revealed that the Mt particle size distribution is polymodal, with small fractions of fine particles (1, 3–7 μm). Particles of 10–40 μm size prevail (about 80%); the peak of the mean diameter is at 21 μm. The diagram of $P_{Rh}$ particle-size distribution is bimodal, with the mean diameter of the major fractions at 45 and 130 μm. In the particle size distribution of the bicomponent system, there are no individual Mt peaks. On the other hand, the mode of the small-size pectin fraction becomes prevalent, its maximum is shifted to 65 μm, and a shoulder at 200 μm appears at the second peak. Such changes suggest the adsorption of Mt particles on the surface of coarser $P_{Rh}$ granules. The interval in which the size characteristic of the fractions increases suggests that the associates incorporate one or several Mt particles. On the other hand, aggregates of $P_{Rh}$ particles with the layered mineral were not found.

The X-ray diffraction patterns shown in Figure 7 illustrate significant changes in the crystal structure of the clay mineral during interaction with the pectin samples.

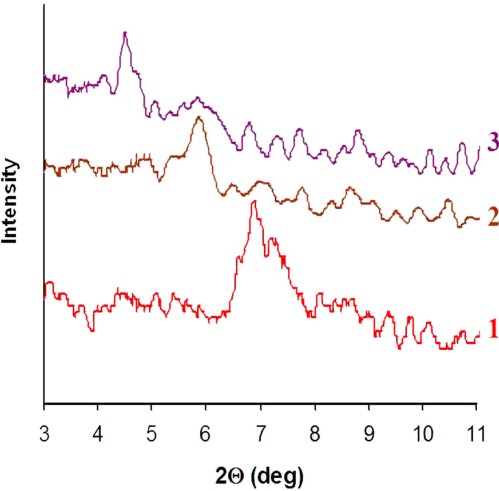

**Figure 7.** X-ray diffraction patterns of (1) initial Mt and bicomponent systems in ratio 90:10; (2) $P_{Ct}$-Mt; (3) $P_{Rh}$-Mt.

In the X-ray diffraction pattern of Mt, there is a basal reflection characteristic of its crystal structure at $2\Theta = 6.9°$. According to [50], this angle corresponds to a distance between the structure-forming layers of 1.28 nm, of which 0.96 nm corresponds to the three-layer aluminum–silicon–oxygen crystal lattice [51] and 0.32 nm to the layer of hydrated ion-exchange cations (gallery).

For bicomponent samples, the diffraction maximum shifts towards smaller angles ($2\Theta \rightarrow 4.5°$). The revealed deviations for the $P_{Rh}$-Mt sample show that the distance between

the Mt structural layers increases to 1.96 nm, which corresponds to an increase in the interlayer space thickness to 1.0 nm. Apparently, this may be due to the intercalation of polyuronide chains into the layered mineral structure. The shift of $2\Theta \to 5.9°$ of the diffraction maximum for the $P_{Ct}$-Mt characterizes the increase in the gallery thickness to 0.6 nm.

This fact gives grounds to consider the $P_{Ct}$–Mt and $P_{Rh}$–Mt bicomponent systems as a composite in which an interfacial layer differing in the structure and properties from the constituent phases is formed. The differences in the intercalation effect are consistent with the tracery of the polymer structure demonstrated in Figure 5. The presence of several *N*-units with a dissociating carboxyl group in the $P_{Rh}$ flexible branches causes the manifestation of electrostatic repulsion forces between the macromolecule sites both in the internal volume of $P_{Rh}$ granules and in fragments located on their surface. The intercalation of such fragments into the swollen Mt structure ensures the fixation of a more expanded interfacial layer than the low-stressed end sections of $P_{Ct}$ macromolecules, after the removal of moisture.

Both in pectin granules and in particles of polymer-inorganic composites, all *N*-units are available for the interaction with theophylline, even those that present as part of crosslinking blocks. There are, at least, no steric hindrances to interaction in each paired repeat of the *C*- and *N*-forms. In a moderately acidic medium, in addition to ionic interactions, the position of the sorbate is maintained by hydrogen bonding with a hydroxyl of the neighboring galacturonate unit (see Scheme 4a). The dissociation of carboxyls is inhibited in a strongly acidic medium, but interaction can occur through hydrogen bonding (see Scheme 4b).

**Scheme 4.** Type of intermolecular interactions during theophylline sorption by pectin: (**a**) pH 5; (**b**) pH 2 (*C*–form is marked by a red color; *N*–form is marked by a brown color; ionic interactions and hydrogen bonds are marked by a magenta color).

The theophylline sorption was studied at a pH corresponding to the physiological norm of acidity in the stomach (pH 2) and in the duodenum (pH 5). A typical view of the kinetic dependences of sorbate absorption by pectin samples at a temperature of 40 °C is shown in Figure 8.

The analysis of empirical data was carried out using Lagergren's (pseudo-first-order) and Ho–McKay's (pseudo-second-order) kinetic models [52–54]. Both models assume that the sorption process is limited by the stage of interaction of the sorbed substance with the adsorption centers. The first model correctly describes the sorption processes preceded by the diffusion stage. The second model is used to describe the processes that include the chemical interaction of the sorbate with the functional groups of the sorbent. The choice of the model was carried out, taking into account the required linear approximation of kinetic dependencies by the following correlation relations:

$$\text{pseudo-first-order model}: \ln(q_e - q_t) = \ln q_e^* - k_1 t$$

$$\text{pseudo-second-order model}: t/q_t = 1/(k_2 q_e^{*2}) + t/q_e^*$$

where $k_1$ is the rate constant of pseudo-first-order absorption (min$^{-1}$) and $k_2$ is the rate constant of pseudo-second-order absorption (g·mmol$^{-1}$·min$^{-1}$); $q_t$ and $q_e$ are experimental values of current and equilibrium sorption (mmol·g$^{-1}$); $q_e$* is the calculated value of the limit sorption capacity of the material (mmol·g$^{-1}$).

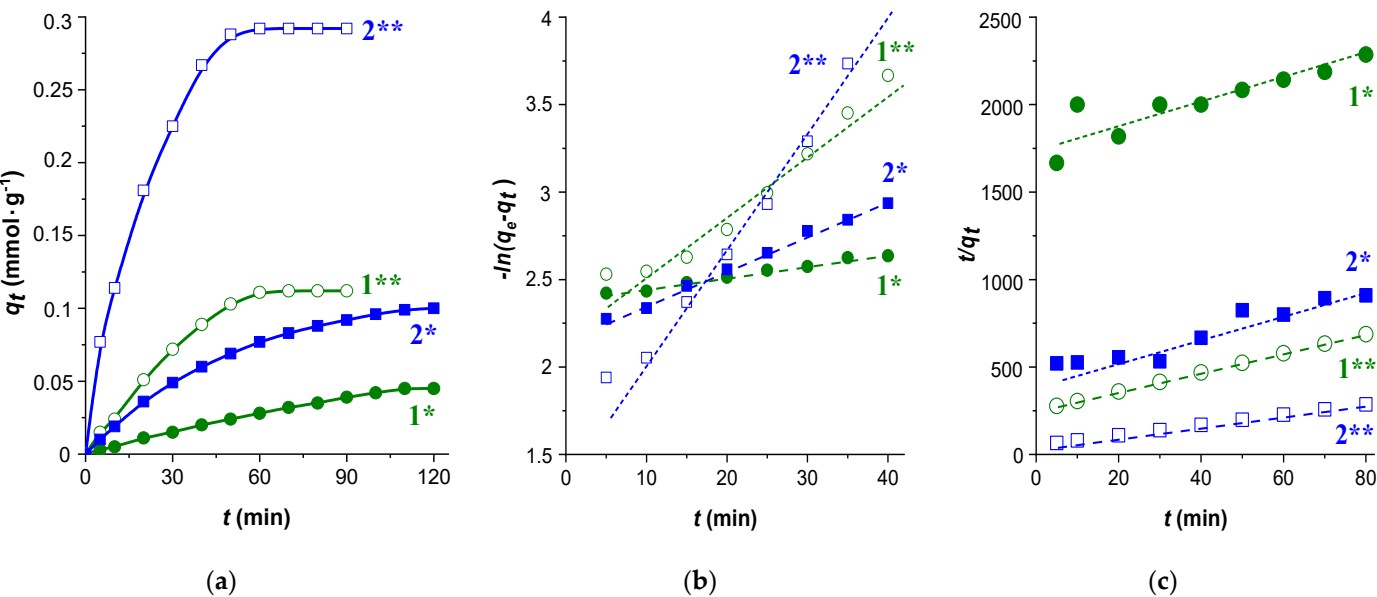

(a)  (b)  (c)

**Figure 8.** (**a**) Kinetics of theophylline adsorption by pectin samples; (**b**) pseudo-first-order model; (**c**) pseudo-second-order models: 1 * and 1 **—P$_{Ct}$, pH 2.0 and 5.0; 2 * and 2 **—P$_{Rh}$, pH 2.0 and 5.0.

Traditionally, the adequacy of the empirical data description is demonstrated graphically in the coordinates ln($q_e - q_t$) versus $t$ for Lagergren's model (Figure 8b) and in the coordinates $t/q_t$ versus $t$ for Ho–McKay's model (Figure 8c). The linear interpretation of the kinetic section of sorption curves in the strongly acidic medium was obtained for the pectin samples in the coordinates of the pseudo-first-order model; for pH 5 it was carried out in the coordinates of the pseudo-second-order model. In the first case, the value of the $k_1$ constant was calculated as the tangent of the inclination angle of the approximating dependence, and $q_e$* was determined by linear extrapolation to the initial moment of the sorption process. In the second variant, the value of the $q_e$* was calculated as the tangent of the slope angle of the linear dependence. The free term was used to calculate $k_2$.

Table 6 compares the parameters of the kinetic models obtained for the sorption curves shown in Figure 8. The inadequacy of the theophylline sorption description by Ho–McKay's model at pH 2 as well as Lagergren's model at pH 5 was expressed, first of all in low values of the determination coefficient, and secondly in considerable deviations of the calculated values $q_e$* from the actual registered values of equilibrium sorption $q_e$. At that, Ho–McKay's model describes the experiment results at pH 5 with an accuracy of over 99% and deviations between the calculated and actual values of the sorption capacity not exceeding 2%. The same high results of sorption data approximation at pH 2 were obtained when Lagergren's model was used.

The comparison of the theophylline sorption rate by pectin preparations can only be carried out within the framework of an appropriate model for a certain pH value. The P$_{Rh}$ sample at pH 5 is better than the commercial analog by 1.44 times (see values $k_2$) and at pH 2 by 3.17 times (see values $k_1$). The comparison of the values of the limit sorption capacity $q_e$* is possible using adequate models both between pectin preparations and when varying the pH. The $q_e$* increases with the increasing of the hydrogen index for the P$_{Ct}$ to 2.37 times and for P$_{Rh}$ to 3 times. At that, the main advantage of sorption at pH 5 is to ensure strong chemosorption binding, which is necessary to prevent the absorption of mycotoxins into the walls of the small intestine.

**Table 6.** Kinetic parameters of theophylline sorption by pectin preparations at 40 °C.

| Pectin | $q_e$ (mmol·g$^{-1}$) | pH | Pseudo-First-Order Model | | | Pseudo-Second-Order Model | | |
|---|---|---|---|---|---|---|---|---|
| | | | $q_e$* (mmol·g$^{-1}$) | $k_1$ (min$^{-1}$) | $R^2$ | $q_e$* (mmol g$^{-1}$) | $k_2$ (g·mmol$^{-1}$·min$^{-1}$) | $R^2$ |
| $P_{Ct}$ | 0.045 | 2 | 0.049 | 0.006 | 0.997 | *0.030* | *0.067* | *0.663* |
| | 0.112 | 5 | *0.118* | *0.052* | *0.803* | 0.116 | 0.119 | 0.992 |
| $P_{Rh}$ | 0.100 | 2 | 0.103 | 0.019 | 0.994 | *0.106* | *0.096* | *0.769* |
| | 0.292 | 5 | *0.364* | *0.071* | *0.887* | 0.309 | 0.171 | 0.995 |

Calculated indicators are italicized for variants that do not meet the conditions for an adequate description of experimental data: $R^2 > 0.9$.

The availability of unsubstituted galacturonate units in the pectin microgranule volume significantly depends on their joint structural distribution with methoxylated and calcium–pectate monomer units. The relationship between the limit sorption capacity of pectin substrates and their chemical structure is described by an equation that is consistent with the dependencies proposed in [31,32]:

$$q_e^* = 0.0887 + 1.1289 \cdot N - 0.1022 \cdot C - 0.1417 \cdot M; \;\; r = 0.9245$$

The model demonstrates that the **M-** and **C**-units and themselves do not participate in the absorption of theophylline, and can also create obstacles to interaction with **N**-units. The obtained equation can be used to predict the preventive effect of plant sorbents on alkaloids using data such as the mass fraction of pectin substances and the ratio of galacturonate units in unsubstituted, methoxylated and calcium–pectate forms in the polymer chain.

The effectiveness of the pectin substrate's action was compared with the results of theophylline sorption by Mt and hybrid composites containing 10% of Mt (Figure 9a).

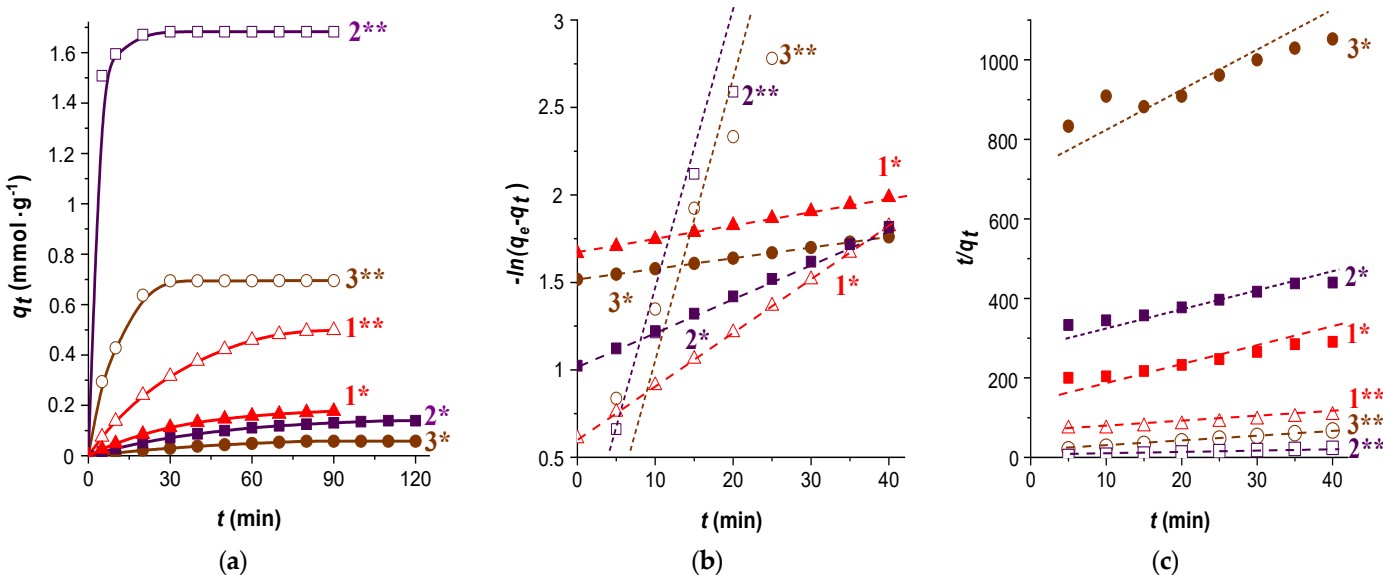

**Figure 9.** (**a**) Kinetics of theophylline adsorption by montmorillonite and bicomponent samples; (**b**) description by pseudo-first-order models; (**c**) description by pseudo-second-order models. 1 * and 1 ** are Mt at pH 2.0 and 5.0; 2 * and 2 ** are $P_{Rh}$–Mt (90:10) at pH 2.0 and 5.0; 3 * and 3 ** are $P_{Ct}$–Mt (90:10) at pH 2.0 and 5.0.

The comparison of the results with Figure 8a data shows that Mt exhibits low sorption properties in a highly acidic medium. However, by the time t = 120 min (the time the food

stays in the stomach [55]), the achieved level $q_{120}$ exceeds the value of the indicator for the preparations $P_{Rh}$ and $P_{Ct}$ by 1.8 and 3.9 times, respectively.

Mt increases sorption activity when acidity decreases. At pH 5, $q_e$ exceeds the value of the equilibrium theophylline adsorption by $P_{Rh}$ and $P_{Ct}$ preparations by 1.7 and 4.4 times, respectively. However, the theophylline molecules are deprotonated in the alkaline medium of the small intestine and lose their connection with the mineral sorbent, while pectins retain the ability of sorption binding of alkaloids by hydrogen bonding.

Figure 9b,c indicate that the pseudo-first-order model provides the best description adequacy for Mt. The kinetics of theophylline sorption by bicomponent preparations at pH 2 are adequately described using Lagergren's model, and at pH 5 are described by Ho–McKay's model. The kinetic characteristics are compared in Table 7.

**Table 7.** Kinetic parameters of theophylline sorption by Mt and bicomponent samples.

| Sorbent | $q_e$ (mmol·g$^{-1}$) | pH | Pseudo-First-Order Model | | | Pseudo-Second-Order Model | | |
|---|---|---|---|---|---|---|---|---|
| | | | $q_e$* (mmol·g$^{-1}$) | $k_1$ (min$^{-1}$) | $R^2$ | $q_e$* (mmol·g$^{-1}$) | $k_2$ (g·mmol$^{-1}$·min$^{-1}$) | $R^2$ |
| Mt | 0.177 | 2 | 0.181 | 0.008 | 0.996 | *0.285* | *0.008* | *0.824* |
| | 0.498 | 5 | 0.504 | 0.030 | 0.992 | *0.862* | *0.022* | *0.841* |
| $P_{Ct}$–Mt (90:10) | 0.058 | 2 | 0.062 | 0.009 | 0.992 | – | – | – |
| | 0.695 | 5 | – | – | – | 0.704 | 0.137 | 0.997 |
| $P_{Rh}$–Mt (90:10) | 0.139 | 2 | 0.141 | 0.036 | 0.995 | – | – | – |
| | 1.685 | 5 | – | – | – | 1.690 | 0.480 | 0.994 |

Calculated indicators are italicized for variants that do not meet the conditions for an adequate description of experimental data: $R^2 > 0.9$.

The results show that hybrid composites are inferior to the Mt preparation at pH 2 in value $q_e$*, but exceed it in absorption rate (see values $k_1$). The sorption rate of bicomponent samples at pH 5 also exceeds the values of $k_2$ constant for pectin samples. At that, the increase of $k_2$ is only 15% for $P_{Ct}$–Mt and reaches 2.8 times for $P_{Rh}$–Mt.

This result is associated with a more significant increase in the interplanar spacing of the layered aluminosilicate upon the intercalation of $P_{Rh}$ particles into the Mt structure. It is important that the space between the Mt layers is a concentrator which increases the sorbate content in the external environment for the pectin component (see Figure 7). At that, the space between the Mt layers is a kind of concentrator that increases the sorbate content in the external environment for the pectin component, which intensifies the theophylline chemisorption binding by the adsorption centers of $P_{Rh}$. As a consequence, the achieved value of $q_e$* index for the $P_{Rh}$–Mt hybrid material exceeds the levels of the individual sorption capacity of the mineral and polymer components by 3.35 and 5.5 times, respectively.

Experiments simulating the passage of the stomach at pH 2 for 120 min, followed by an increase in the hydrogen index to pH 5, were also carried out. The results of the analysis of sorption curves are presented in Table 8.

As can be seen from the obtained data, the polymer and composite materials under study are not passive at the stage of treatment in a strongly acidic solution of theophylline. At the same time, the $q_e$* for each sorbents remains the same as that without pretreatment. Apparently, the determining factors in this case are the same degree of adsorption centers activation when the level of the hydrogen index increases to pH 5, and the associated changes in the structural state of pectin grains and hybrid intercalated systems.

**Table 8.** Kinetic parameters of theophylline binding by pectin and bicomponent samples at pH 5 after pre-sorption at pH 2 for 120 min.

| Sorbent | $q_e$ (mmol·g$^{-1}$) | Pseudo-Second-Order Model | | |
|---|---|---|---|---|
| | | $q_e$* (mmol·g$^{-1}$) | $k_2$ (g·mmol$^{-1}$·min$^{-1}$) | $R^2$ |
| P$_{Ct}$ | 0.115 | 0.116 | 0.150 | 0.995 |
| P$_{Rh}$ | 0.305 | 0.309 | 0.315 | 0.991 |
| P$_{Ct}$–Mt (90:10) | 0.701 | 0.704 | 0.137 | 0.997 |
| P$_{Rh}$–Mt (90:10) | 1.689 | 1.690 | 0.965 | 0.994 |

The main difference is an increase in the rate of theophylline absorption. The increase with the constant $k_2$ is 1.26 times for a commercial P$_{Ct}$ sample used as a standard for comparison, and is 1.39 times for the hybrid composite P$_{Ct}$–Mt. The P$_{Rh}$ preparation demonstrates a more pronounced significance of theophylline pre-binding by the physical sorption mechanism in a highly acidic medium stomach, followed by chemosorption fixation in the duodenal zone: the preliminary stage provides an increase of the $k_2$ constant to 1.9 times. A twofold increase of the $k_2$ for the P$_{Rh}$–Mt sample indicates the manifestation of the effect of an additional increase in the concentration gradient of theophylline in the surface layer of intercalated Mt. At that, the ratio of $k_2$ for the bicomponent P$_{Rh}$–Mt system and the base pectin P$_{Rh}$ increases up to 3.2 times. The revealed patterns are of fundamental importance, taking into account the limited time period noted above for the effective blocking of mycotoxins in the human body.

The estimated sorption capacity of the pectin–montmorillonite coating onto the cellulose matrix of the modified Rh$^{bio}$–Mt phytopreparation can be calculated if the $q_e$* value for the P$_{Rh}$–Mt hybrid material is expressed in weight fractions of absorbed theophylline, and the pectin content in the Rh biomass is taken into account. The estimated value of the indicator is 38.03 mg·g$^{-1}$. The obtained value is comparable with the experimental value of the equilibrium sorption of the model alkaloid by Rh$^{bio}$–Mt (see Table 3). Thus, the sorption activity of the modified phytopreparation is almost completely determined by the properties of the hybrid polymer-inorganic coating.

The obtained results have practical importance. The values of the absorption rate constants and the maximum sorption capacity allow the determination of the dosage of pectin-containing phytopreparations for the prevention of mycotoxicosis from the calculation of the specific binding of alkaloids. For this, the integral equation of the pseudo-second-order adsorption model can be used:

(a)    During the passage of food in the human duodenum:

$$q_{20} = 20 \left/ \left( \frac{1}{k_2 q_e^{*2}} + \frac{20}{q_e^*} \right) \right.$$

(b)    During the passage of feed mass in the body of farm animals:

$$q_{30} = 30 \left/ \left( \frac{1}{k_2 q_e^{*2}} + \frac{30}{q_e^*} \right) \right.$$

## 4. Conclusions

The presented results indicate that our proposed technology for modifying rhubarb biomass by controlled enzyme action on cellulose matrix polymers and the solid-phase synthesis of a hybrid pectin–montmorillonite coating enhances the functionality of phytosorbents for the effective binding of azaheterocyclic mycotoxins in the digestive systems of humans and farm animals. Confirmation of the super-additive influence of the components in the development of porosity and sorption properties of the hybrid composite

material was obtained. The effect of the pectin supramolecular structure on the intercalation of the layered mineral was traced. Simulation of the block-cellular structure of rhubarb pectin was carried out according to the degree of polymerization and the ratio of the galacturonate unit forms in comparison with the commercial citrus pectin characteristics. The relationship between the structural differences of pectins and kinetic and equilibrium parameters of theophylline sorption was traced. The rate constants of absorption and the ultimate sorption capacity $q_e*$ of pectin and hybrid substrates at pH 2 and pH 5 were analyzed using the pseudo-first- and pseudo-second-order kinetic models. A correlation model was obtained for predicting the effects of the level of $q_e*$ in pectin sorbents to alkaloids according to the ratio data of the galacturonate units forms. It was shown that intercalated montmorillonite performs a concentrator function which increases the sorbate content in the external environment to intensify the theophylline chemosorption binding on the adsorption centers of pectin. The practical importance of the obtained values of the adsorption rate constant and the $q_e*$ index is associated with the possibility of justifying the dosage of modified enterosorbents for the effective immobilization of azaheterocyclic mycotoxins for a limited time of passage in the duodenal of humans and farm animals.

## 5. Patents

Patent RU 2666769 C1. Method for fodder production from plant raw materials with a high content of lignified fiber. Publ. 12 September 2018.

**Author Contributions:** Conceptualization, N.K. and S.K.; methodology, S.A. and O.L.; validation, O.L., A.B., E.N. and N.K.; formal analysis, O.L. and S.A.; writing—original draft preparation, N.K. and S.K.; writing—review and editing, A.B.; visualization, O.L. and S.A.; supervision, A.B.; project administration, E.N. All authors have read and agreed to the published version of the manuscript.

**Funding:** This work was financially supported by the Ministry of Science and Higher Education of the Russian Federation (state contract No. 122040500050-5).

**Institutional Review Board Statement:** Not applicable.

**Informed Consent Statement:** Not applicable.

**Data Availability Statement:** Data are contained within the article.

**Acknowledgments:** This work was performed on equipment at the shared resource center of the Upper Volga Regional Center of Physicochemical Studies.

**Conflicts of Interest:** The authors declare no conflict of interest.

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
