# Peer review of "Enterosorbents Based on Rhubarb Biomass with a Hybrid Polymer-Inorganic Coating for the Immobilization of Azaheterocyclic Mycotoxins"

_coatings, doi:10.3390/coatings13040684_

Round 1

Reviewer 1 Report

The manuscript entitled "Enterosorbents Based on Rhubarb Biomass with a Hybrid  Polymer-Inorganic Coating for the Immobilization of Azahet-erocyclic Mycotoxins" showed an exciting way to use the Rhubarb biomass.

Some suggestions as follows:  

1.       For the Materials and Methods, I suggest making it with a sub-title to make it easy to follow.  (such as,,,, 2.1 Materials; 2.2 Rhéum rhabarbarum preparation; 2.3 evaluation of the Rh–Mt and Rhbio–Mt structure…etc)

2.       Line 166-167 Please provide more information about this sentence.

3.       Line 124-130 I suggest making it a table in order to make it easy to follow.

Line 269-276 needs to support the sentences with previous work.

5.       Line 300-301, this sentence, "The system sorption properties are almost fully realized within 20 minutes which are necessary to effective mycotoxins binding into the human body", need more explanation of what evidence you have to mention that 20 min necessary to effective mycotoxins binding into the human body.

6.       Line 330-340 need to be supported by references.

7.       Line 357-358 is not clear. What do you mean?

8.       Line 554-557, 574-581 need to be supported by references

Regards,

Author Response

The authors agree with the reviewer's comments and we are grateful for the useful comments. The corrections were made in manuscript.

Reviewer 2 Report

Dear,

The authors evaluated the effectiveness of biochemical modification of plant raw materials to form a pectin-montmorillonite coating with active absorption. The manuscript is a good contribution to the literature and therefore has merit for publication. Some small details:

> Page 2. Introduction. Please add Figure caption and numbering in the introduction;

> Introduction. Please make clear the innovation and novelty of the manuscript, specifically the literature gap;

> Figure 2. Did the authors not perform triplicates? Experimental error must be reported;

> Authors should report articles from the literature to support discussions;

Author Response

(The authors gave the same response as above.)

Round 2

Reviewer 1 Report

Dear Authors,

Thank you for your revision. 

Regards,